# Evaluating the Service Performance of Heavy Axle Load Ballasted Railway by Using Numerical Simulation Method

Shuang Tian [1,2,3,*], Xianzhang Ling [1,3], Ting Li [2], Andrew Chan [4] and Ionut-Razvan Georgescu [1]

1   School of Civil Engineering, Harbin Institute of Technology, Harbin 150090, China; lingxianzhang@hit.edu.cn (X.L.); i.georgescurazvan90@yahoo.com (I.-R.G.)
2   State Key Laboratory for Geomechanics and Deep Underground Engineering, China University of Mining and Technology, Xuzhou 221008, China; liting@cumt.edu.cn
3   Chongqing Research Institute, Harbin Institute of Technology, Chongqing 401135, China
4   School of Engineering, University of Tasmania, Hobart 7001, Australia; andrew.chan@utas.edu.au
*   Correspondence: ts_hit@163.com

**Abstract:** To evaluate the service performance of the track substructure of heavy axle load (HAL) railway transportation, an inverse analysis was performed to estimate the resilient modulus values of the track substructure, based on the deflection data obtained from light falling weight deflectometer testing. Subsequently, a three-dimensional finite element model was developed to simulate the effect of the train speeds ($v$) and axle loads ($F$) on the typical dynamic responses in the railway track system. The results convincingly indicated that increasing $v$ or $F$ can amplify the track vibration. Finally, a critical stress ratio method was adopted to evaluate the service performance based on the numerical results. A recommended range of $v$ and $F$ was determined to maintain the long-term stability of the HAL railway line. The findings can provide guidance for designing the track and maintenance plans to avoid track support failures and ensure track infrastructure resiliency.

**Keywords:** service performance evaluation; empirical equation; dynamic response; light falling weight deflectometer; finite element method; heavy axle load railway



## 1. Introduction

The rapid growth of the world's freight transportation network has accelerated the development of heavy axle load (HAL) railway transportation. The HAL railway system, which is influenced by dynamic train loads, can adversely impact the reliability of the roadbed, owing to the high amplitude and frequency of the associated loading cycles. Therefore, it is necessary to evaluate whether the existing HAL railway tracks can satisfy the requirements to support trains with a higher speed and larger axle load to improve the transportation capacity of the HAL railway system [1].

In particular, railway tracks can exhibit a deteriorated performance after being subjected to large axle loads or high train speeds, thereby failing to provide the required stability [2]. The service performance of tracks is generally evaluated considering the dynamic response caused by moving trains [3], and the dynamic behavior of the railway system can be examined using numerical simulations. In this regard, the train speed ($v$) and axle load ($F$) have been identified as the main factors that contribute to the dynamic response of HAL railways [4,5]. Several numerical models for the railway track have been established to determine the effect of $v$ and $F$ on the dynamic responses [5–8]. However, in these studies, the combined effect of $v$ and $F$ was not considered; that is, the results were limited to a single factor ($v$ or $F$). Therefore, it is desirable to develop a united empirical equation of the dynamic response, taking into account the numerical results, which can be used to predict the acceleration and dynamic stress under the combined effect of $v$ and $F$.

Furthermore, the abovementioned numerical models usually require elastic parameters (i.e., resilient modulus and Poisson's ratio) to be determined for each layer of the

substructure. They are both usually obtained by conducting laboratory tests or in situ measurements under those of the loading environment [9,10]. However, collecting soil samples is often not feasible owing to cost and time limitations and the occurrence of traffic disruption. Moreover, soil samples alone cannot sufficiently indicate the track performance [11–13]. A method to efficiently and accurately determine the modulus is being considered as the foundation for further development of railway engineering and thus has considerable theoretical and practical significance. One measurement technique, originally developed for road and airfield pavements, involves the use of a light falling weight deflectometer (LFWD), which has been widely applied to investigate the track substructure in the United Kingdom, Ireland, Germany, USA, and Canada [13–18]. However, a complete description pertaining to the estimation of the resilient modulus of the HAL track substructure in China has not been provided yet.

Analyzing the deflection data collected using the LFWD is a prompt and reliable approach to characterizing the properties of the subgrade layers, and extensive efforts have been made to discuss the use of these data as a performance measure for the railway behavior [19–22]. In addition, several methods to evaluate the performance of railway crossing rails have been presented [23,24]. However, most of these research efforts were directed toward understanding the effects of the materials, moisture, and seasonal variation. Until now, the problems of the service performance evaluation of HAL railways under the dynamic impact of HAL trains have not been focused on. Therefore, it is necessary to evaluate the stability of HAL railways considering an increase in the v and F of HAL trains.

To this end, this paper presents an approach involving LFWD testing to estimate the resilient modulus of the track substructure. In particular, a finite element (FE) model is established using ABAQUS software, in which an inverse analysis subroutine is used to estimate the subgrade modulus from LFWD field test measurements. Herein, a user subroutine within ABAQUS (i.e., UMAT, user-defined materials) was used to present the nonlinear of the subgrade. To demonstrate the approach, a case study is performed using data obtained from an existing HAL railway line in northwest China, which runs between Baotou and Shenmu. Furthermore, a 3D FE model is established to investigate the HAL-train-induced vibration behavior of the track, and the model was validated via field testing. It should be noted that the ABAQUS software is used to conduct the inverse and dynamic analysis in this research. In addition, certain empirical equations for the typical dynamic response are formulated considering the combined effect of $v$ and $F$, and the method of service performance evaluation is discussed. The critical stress ratio is adopted as an indicator to assess the long-term stability of a HAL railway line under suitable $v$ and $F$.

## 2. Inverse Analysis Pertaining to LFWD Test

### 2.1. Overview of LFWD Test and Its Inverse Analysis

The section DK65 + 629 on the Baotou–Shenmu HAL railway is selected to carry out the LFWD field test (Figure 1) [25,26]. As shown in Figure 2a, the LFWD used in this study is developed and manufactured by the Rincent BTP company. The operation mode of the device is simple. As the sliding drop weight is released, it strikes the cylindrical shock absorbers so that an impulse load of 40 ms duration with a maximum force ranging up to 35 kN is transferred through the loading plate into the ground [27]. The force is measured using a load cell on the center of the plate, and geophones are used to measure the surface accelerations at various distances from the footplate [14]. The vertical displacements ($d_n$) can be then obtained by integrating the accelerations. On this basis, the surface deflection at different points away from the load application point can be then determined.

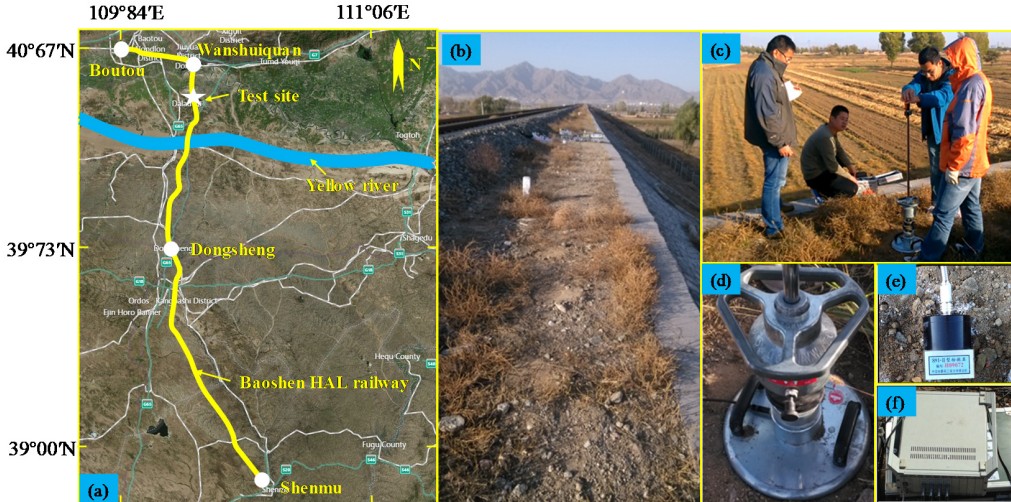

**Figure 1.** LFWD test plan: (**a**) location of the Baoshen heavy axle load (HAL) railway; (**b**) test site; (**c**) LFWD test; (**d**) LFWD device; (**e**) vibration pickup (891-II); and (**f**) data acquisition system (DH5922D).

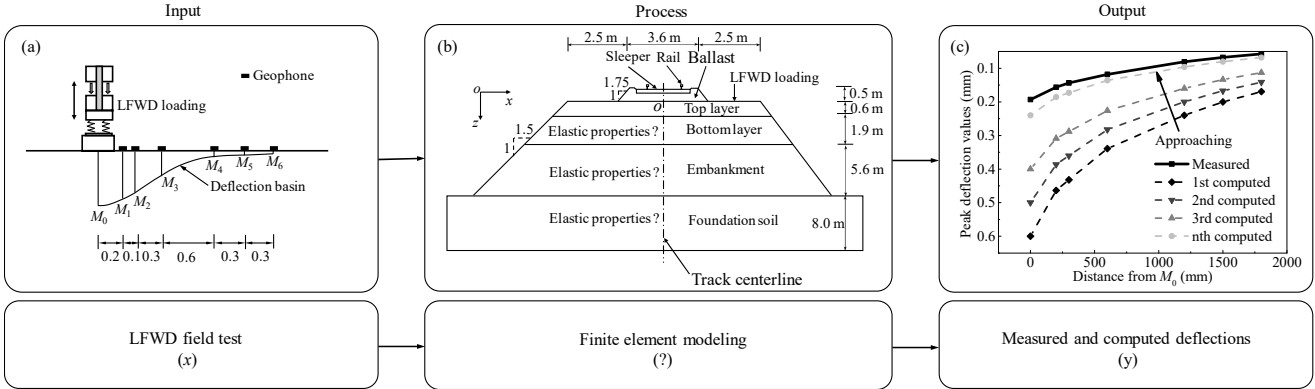

**Figure 2.** Inverse problem (**a**) input: LFWD field test; (**b**) process: finite element modeling; and (**c**) output: comparison of measured and computed deflections.

The geophone distances were selected to target the displacement responses that are a result of, and representative of, the differing layers of material that comprise a railway track bed (see Figure 2a). They were based on previous works that empirically relate the depth of materials affected by the load to the displacement responses measured at differing distances from the applied load [14,18,28]. The geophones at 0.2, 0.3, and 0.6 m ($M_1$–$M_3$) provide a combined measure of the whole of the support structure; and the 1.2, 1.5, and 1.8 m ($M_4$–$M_6$) geophones provide a measure of the combined stiffness of the embankment and subgrade. The depth of materials varies over the length of the railway structures and, as a result, the geophone placements are rarely optimal; however, experience has shown these placements to be practical [14,18,28].

Subsequently, the deflection data resulting from an LFWD test can be used in combination with a model of the railway track structure to determine the values of the resilient modulus necessary to characterize the material properties of the track substructure [14,29]. To this end, in the initial stage, the estimated values of the modulus are adopted for each layer of the substructure in the FE modeling (Figure 2b), and the comparison between the resulting deflections from the model and the field is then conducted.

In the end, the influence of the resilient modulus and Poisson's ratio of each layer on the magnitude of the deflections can be determined by performing a trial and error process or a sensitivity analysis (Figure 2c).

### 2.2. Three-Dimensional Numerical Model

The ABAQUS standard software is adopted to develop a three-dimensional FE model to realize the LFWD testing on a railway track (Figure 3). Considering the symmetry of the cross-section, a half-track model is developed. The load is assumed to act at the same position as the test point of $M_0$, and the load pulse is idealized using a half sine function with a duration of 40 ms and an amplitude of 35 kN. It should be noted that the duration and amplitude of the load pulse used in the FE model are consistent with LFWD loading obtained in the field tests. Moreover, it has been demonstrated that the load pulse using a half sine function form is reasonable, following the study of Burrow et al. [14]. A 20-node quadratic brick reduced integration element (C3D20R) is used to represent the ballast, subgrade, and ground elements. In any dynamic analysis, the finite element size has to be selected carefully to ensure the accuracy of results. In general, the element size of the FE model was estimated based on the smallest wavelength that allows the high-frequency motion to be simulated correctly [30]. Accordingly, the sizes of used 3D finite elements were taken as $0.05 \times 0.15 \times 0.1$, $0.075 \times 0.08 \times 0.1$ $0.13 \times 0.11 \times 0.1$, $0.13 \times 0.24 \times 0.1$, and $0.4 \times 0.56 \times 0.1$ for the ballast, top layer, bottom layer, embankment, and foundation, respectively. Hence, the FE mesh contains 235,812 elements and 253,518 nodes.

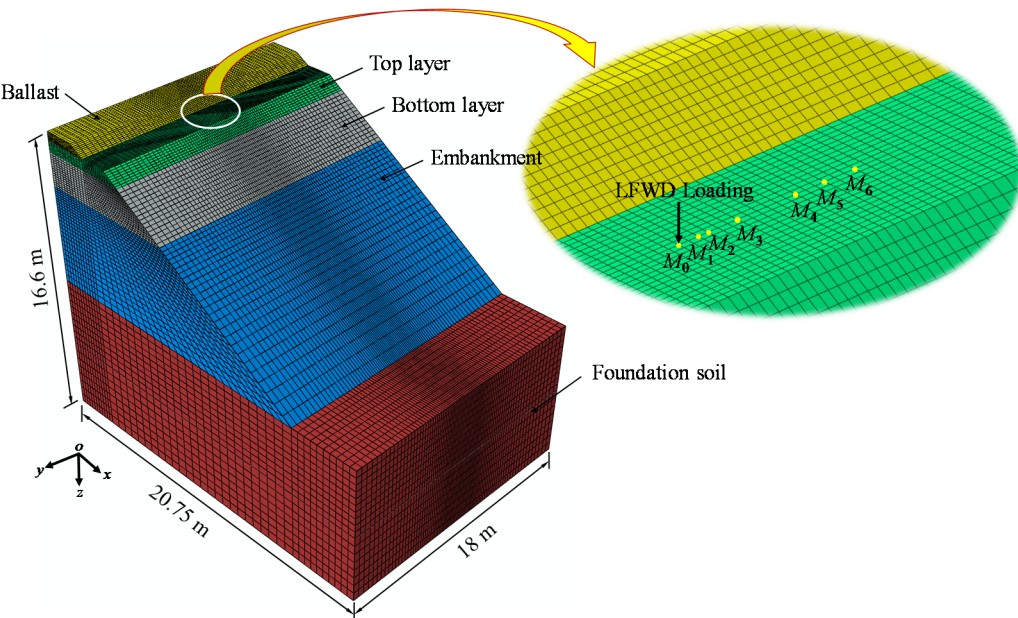

**Figure 3.** Track model for LFWD inverse analysis.

### 2.2.1. Boundary Conditions

In the current case, the LFWD load is a transient dynamic load with a duration of 40 ms [14]. Meanwhile, the shear wave velocity $Vs$ in the top layer, bottom layer, embankment, and foundation can be approximately determined using the equation $(V_s = \sqrt{\frac{E}{2(1+\nu)\rho}})$ as 232.9, 202.4, 196.9, and 150.0 m/s, respectively. Based on the description mentioned above, during the 40 ms period of analysis, a wave will travel approximately 5.6 m away from the load. Therefore, the model is made sufficiently large (18, 20.75, and 16.6 m in the longitudinal ($y$), horizontal ($x$), and vertical ($z$) directions) to ensure that there is insufficient time for the reflected shear waves to return to the part of the model of interest during the analysis period of 40 ms. To this end, the bottom boundaries are restrained (pinned) in the vertical and horizontal directions, and the side boundaries are constrained in the lateral direction only. Moreover, due to half symmetry, the model is established with the symmetrical boundary at the center surface.

### 2.2.2. Track Substructure Properties Required for Structural Analysis

For granular materials, such as the coarse-grained materials used for the subgrade layer, the stress is the primary factor affecting the resilient modulus. It has been noted that the resilient modulus increases considerably with an increase in the confining stress and only slightly with an increase in the repeated deviator stress. In this work, a widely used model for the resilient modulus of granular materials, proposed by Tam and Brown [31], was used to represent the granular material behavior, as shown in Equation (1).

$$E = K_1 \left( \frac{p}{q} \right)^{K_2} \tag{1}$$

where $E$ is the resilient modulus, $p$ is the mean normal compressive stress, $q$ is the deviator, and $K_1$ and $K_2$ are material constants. In the three-dimensional stress state,

$$p = \frac{\sigma_x + \sigma_y + \sigma_z}{3} \tag{2}$$

$$q = \frac{\sqrt{\left[ (\sigma_x - \sigma_y)^2 + (\sigma_y - \sigma_z)^2 + (\sigma_z - \sigma_x) + 6 \left( \tau_{xy}^2 + \tau_{yz}^2 + \tau_{zx}^2 \right) \right]}}{2} \tag{3}$$

where $\sigma_x$, $\sigma_y$, and $\sigma_z$ denote the normal compressive stresses in the $x$, $y$, and $z$ directions respectively, and $\tau_{xy}$, $\tau_{yz}$, and $\tau_{zx}$ represent the shear stresses in the $xy$, $yz$, and $zx$ planes respectively. For the ballast and foundation soil used for the ground, the linear elastic model is adopted for the sake of simplicity.

### 2.3. Result of Inverse Analysis

#### 2.3.1. Analysis

The analysis has three steps, i.e., Step I (when the FE model contains only the ground layers, and an initial geostatic stress field is generated by performing a static analysis by using the ABAQUS initial condition option), Step II (the ballast and subgrade layers are added to the model, and the resulting vertical displacements at the sensors are computed), and Step III (the LFWD load is applied to the subgrade surface, and a time integration dynamic analysis is performed). The time step is estimated through the Courant–Friedrichs–Lewy condition. Because of the high density and elastic modulus of the top layer, $\Delta t$ adopted in this model is primarily controlled by the top layer elements, which are around $5 \times 10^{-4}$ s. The deflection–time histories are computed by subtracting the deflections obtained during Step II from those obtained in Step III.

As shown in Equation (1), the back-calculation parameters are base modulus value ($K_1$) and exponent coefficient ($K_2$). Due to the complexity of Equation (1), it is difficult to obtain the analytic solution to determine the back-calculation parameters. Therefore, the method of trial-and-error was used in this study. In the calculation, the initial back-calculation parameters ($K_1$ and $K_2$) were first set. A base modulus and an exponent coefficient are set by experience from the previous studies [4,14,29], which is also listed in Table 1. Using the initial estimated parameters of each layer, the peak deflections at the sensors are computed and compared with those obtained from the field measurements. Subsequently, the modulus values are modified by trial-and-error until the measured values of the deflections closely match the corresponding values obtained from the simulation. The difference between the calculated and measured LFWD deflections, which is known as the percentage deflection fitting error and can be calculated using Equation (4) [14], must be minimal. In this work, the value of this error is expected to be less than 3%.

$$\varepsilon_{def} = \frac{1}{n} \sum_{i=1}^{n} \frac{|d_{m,i} - d_{c,i}|}{d_{m,i}} \times 100\% \tag{4}$$

where $n$ is the number of sensors, $d_{m,i}$ is the deflection measured using sensor $i$, and $d_{c,i}$ is the calculated deflection for sensor $i$.

**Table 1.** Properties of track component materials.

| | | |
|---|---|---|
| Sleeper | Young's Modulus, *E* (GPa) | 30 |
| | Poisson's ratio | 0.18 |
| | Density (kg/m$^3$) | 2500 |
| | Length × Width × Height (m) | 2.6 × 0.25 × 0.2 |
| Ballast | Young's Modulus, *E* (MPa) | 180 |
| | Poisson's ratio | 0.25 |
| | Density (kg/m$^3$) | 1700 |
| Top layer | [1] Base modulus $K_1$ (MPa) | 275 |
| | [1] Exponent $K_2$ | 0.25 |
| | Poisson's ratio | 0.3 |
| | Density (kg/m$^3$) | 1950 |
| Bottom layer | [1] Base modulus $K_1$ (MPa) | 196.6 |
| | [1] Exponent $K_2$ | 0.2 |
| | Poisson's ratio | 0.25 |
| | Density (kg/m$^3$) | 2000 |
| Embankment | [1] Base modulus $K_1$ (MPa) | 186.1 |
| | [1] Exponent $K_2$ | 0.2 |
| | Poisson's ratio | 0.25 |
| | Density (kg/m$^3$) | 2000 |
| Foundation soil | Young's Modulus, *E* (MPa) | 120 |
| | Poisson's ratio | 0.27 |
| | Density (kg/m$^3$) | 2100 |

[1] Back-analyzed value.

### 2.3.2. Results

The analysis indicates that the FE model can be calibrated for the LFWD deflections at the test site by using the appropriate constitutive equations, given as Equation (1), together with the base modulus and exponent values presented in Table 1. In addition, the ballast and ground are considered as linear elastic materials, and the corresponding parameters adopted in previous research [6] are summarized in Table 1. The constitutive equations can be used to represent the behavior of the subgrade layer, and the LFWD deflections determined from the FE model are shown in Figure 4. The deflection fitting error obtained using Equation (4) is 1.84%, which is less than the considered tolerance of 3% [14].

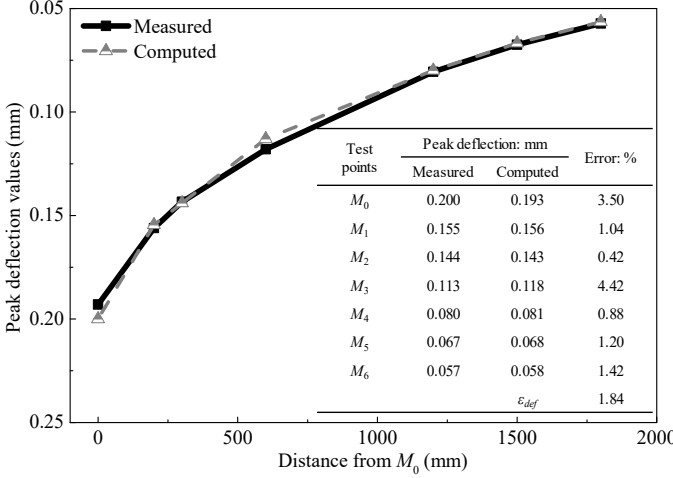

| Test points | Peak deflection: mm | | Error: % |
|---|---|---|---|
| | Measured | Computed | |
| $M_0$ | 0.200 | 0.193 | 3.50 |
| $M_1$ | 0.155 | 0.156 | 1.04 |
| $M_2$ | 0.144 | 0.143 | 0.42 |
| $M_3$ | 0.113 | 0.118 | 4.42 |
| $M_4$ | 0.080 | 0.081 | 0.88 |
| $M_5$ | 0.067 | 0.068 | 1.20 |
| $M_6$ | 0.057 | 0.058 | 1.42 |
| $\varepsilon_{def}$ | | | 1.84 |

**Figure 4.** Comparison of measured and computed peak deflection values at different test points.

## 3. Dynamic Analysis

### 3.1. Three-Dimensional Numerical Model

3.1.1. Track Geometry and Materials

The geometry and subgrade profile of the section DK65 + 629 on the Baotou–Shenmu HAL railway line are shown in Figure 1b, and the 3D FE dynamic model, consisting of 190,078 elements and 201,812 nodes, is shown in Figure 5. The model dimensions were 18, 41.5, and 12.6 m in the longitudinal, horizontal, and vertical directions, respectively. All the track components (i.e., sleeper, ballast, top layer, bottom layer, embankment, and foundation) were modeled using 3D solid elements. To represent model geometry, 25 sleepers were placed along the longitudinal direction at spacing intervals of 0.6 m. The interaction between sleepers and ballast is set to be "hard contact" together with a friction of "rough". The sleepers, ballast, and foundation soils were considered as linear elastic materials, whereas the top layer, bottom layer, and embankment were modeled using user-defined materials, as expressed in Equation (1). The material constants were obtained based on the LFWD inverse analysis described previously. The properties of all the materials considered for this model are summarized in Table 1.

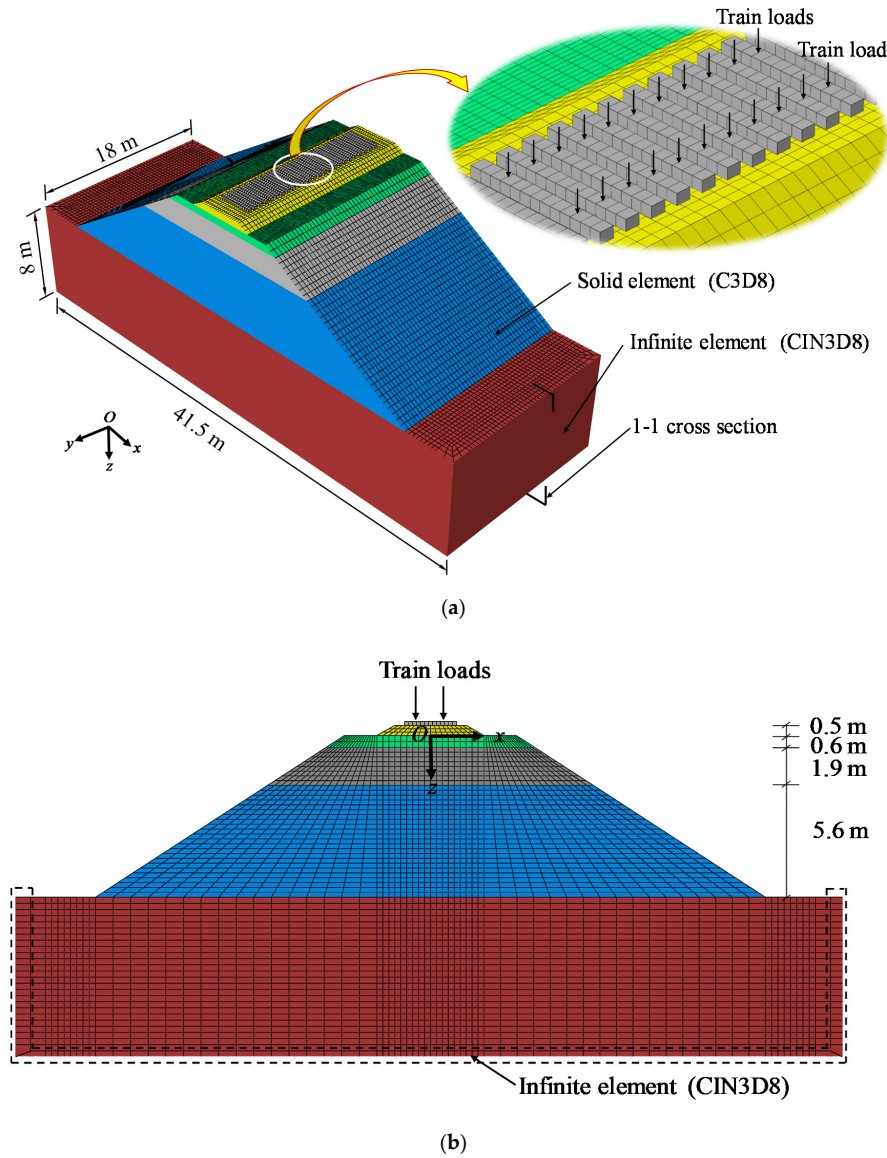

**(a)**

**(b)**

**Figure 5.** Track model for dynamic analysis: (**a**) complete model and (**b**) 1-1 cross-section.

In a dynamic analysis, the model boundaries must be selected carefully to ensure the accuracy of the results. Furthermore, as described in Section 2.2.1, it is necessary to simulate the effect of an unbounded domain for the case of long-term periodic or moving loads. Hence, the FE meshes were surrounded by the infinite elements at the bottom and side boundaries to absorb the stress waves at the boundary surfaces (see Figure 5b).

### 3.1.2. Train Loads

In the coupling calculation of the train–track system, the iteration convergence of the two subsystems is required, which is extremely complicated and time-consuming [32]. Therefore, in this work, the train loads were calculated first and then applied to the rail supporting nodes of the sleeper beam elements (Figure 5a).

To derive the train loads, a series of moving axle loads can be obtained according to the train geometry shown in Figure 6a. Accordingly, the successive axle loads can be expressed as a periodic function $f(t)$ with a period of $t_4$ [30]:

$$f(t) = \begin{cases} F & 0 \le t \le t_1 \\ 0 & t_1 \le t \le t_2 \\ F & t_2 \le t \le t_3 \\ 0 & t_3 \le t \le t_4 \end{cases} \tag{5}$$

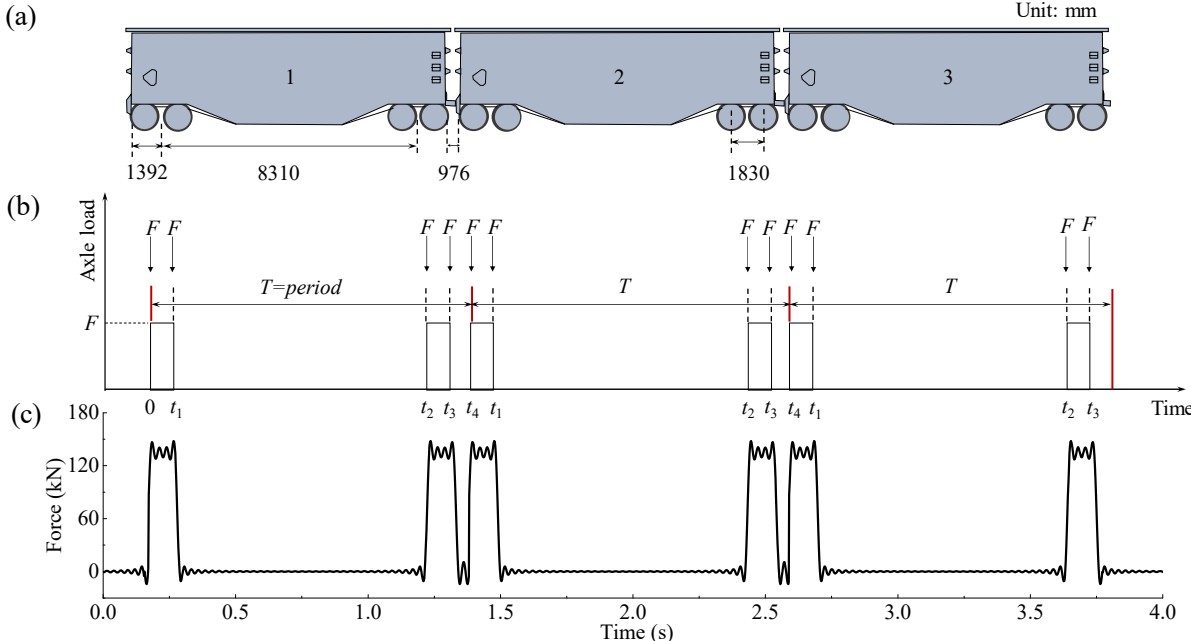

**Figure 6.** Schematic of (**a**) train wagons, (**b**) normal force variation of wagon wheels, and (**c**) dynamic load of a train with an axle load of 275 kN and speed of 60 km/h.

For the sake of simplicity, $f(t)$ is supposed as a sufficiently smooth function, which can then be transformed using the Fourier function, which is composed of a series of sines and cosines.

$$f(t) = a_0 + \sum_{n=1}^{\infty} \left[ a_n \cos\left(\frac{2n\pi t}{T}\right) + b_n \sin\left(\frac{2n\pi t}{T}\right) \right] \tag{6}$$

where $a_0$, $a_n$, and $b_n$ are the Fourier series coefficients. Therefore, the train loads can be calculated if $a_0$, $a_n$, and $b_n$ are determined. It should be noted that the principal frequencies in Equation (6) range from 0 to $\infty$. However, according to Hamid et al. [33], considering the frequency range from 0 to 25 Hz is sufficient to accurately calculate the train loads. The adoption of this range of frequency is dependent on the previous study, which has

shown that the dominant frequencies were found within 10 Hz [25]. Therefore, in this study, only the first 46 terms of Fourier series that have a significant energy were considered, and loads with frequencies greater than 25 Hz were ignored. Figure 6c shows the dynamic load of a train with a wagon length of 11.094 m, an axle load of 275 kN, and a speed of 60 km/h.

The train loads calculated using the method described above were applied to the sleepers with a shift in the loading time to simulate the passage of a train bogie. At a speed of 60 km/h, for example, the train requires 0.036 s to traverse a distance of 0.6 m; thus, 0.036 s is the loading interval in the time domain. This method of loading has been used in the previous study and demonstrated to be correct and reasonable [10].

### 3.2. Validation Using Experimental Data

The model performance was analyzed by comparing the vibration results to the field trials collected in the Baoshen HAL railway between Baotou and Shenmu [25]. Figure 7 shows a comparison between simulation results and field experiments. Predicted soil response has a high correlation of peak particle acceleration with the experimental result. Figure 8 shows the attenuation of the vertical peak acceleration ($A_{max}$) with an increase in the distance from the track centerline. Strong similarities can be noted between the magnitudes and gradients of both the lines, and the correlation between the results is 0.98, which was rendered as sufficiently accurate by Connolly et al. [34]. Therefore, the numerical model can be considered to be capable of effectively simulating the $A_{max}$ both near and far from the track, which demonstrates the reliability of the calculation method adopted in this work.

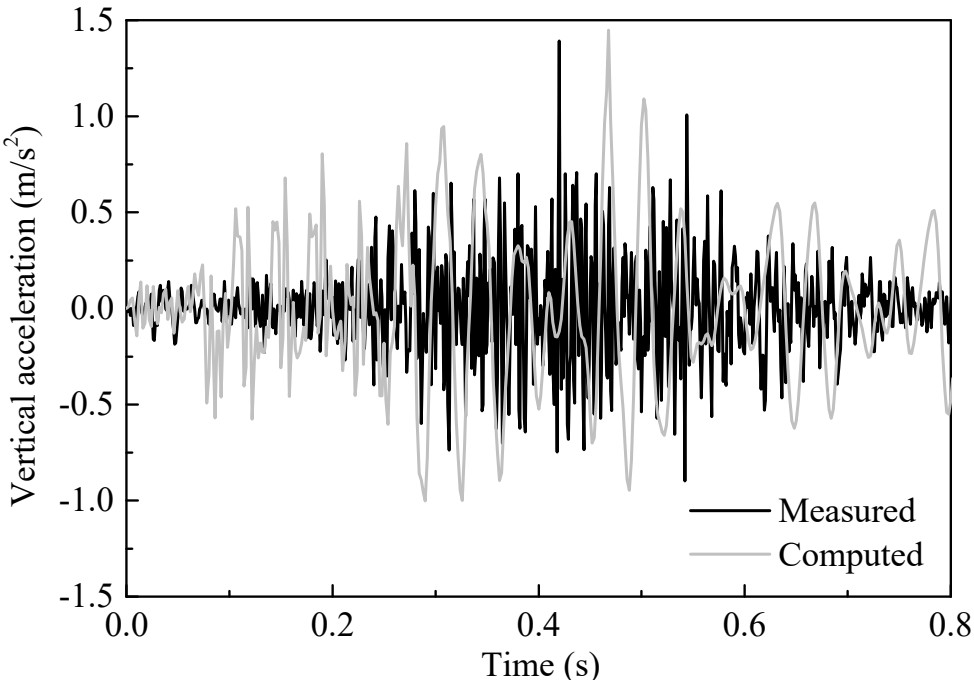

**Figure 7.** Computed and measured acceleration time histories on the top layer surface.

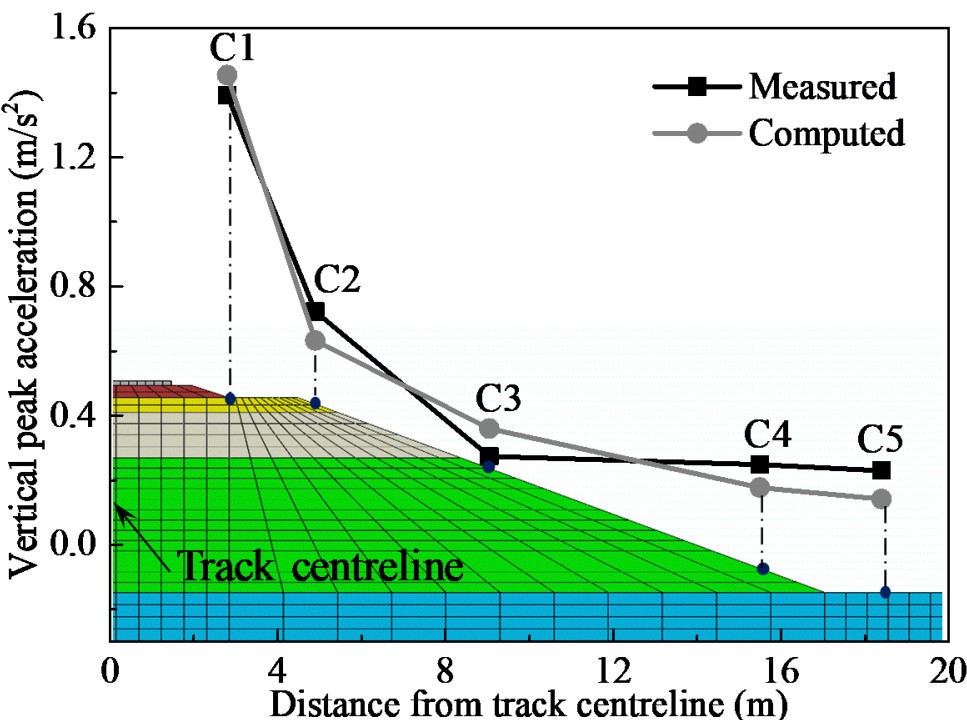

**Figure 8.** Comparison of peak accelerations values of the subgrade with the distance from the track centerline.

### 3.3. Result of Dynamic Analyses

Using the validated computation model, a parametric study was performed to evaluate the influence of *v* and *F* on the predicted dynamic response of the reference service performance. The dynamic response of the reference track model was evaluated based on the vertical peak values of the acceleration and dynamic stress observed within the track.

#### 3.3.1. Effect of Train Speed

As a representative example, Figure 9 shows the resulting vertical peak values of the acceleration and dynamic stress observed at the center of the top surface layer. As expected, the acceleration and dynamic stress both tend to increase with an increase in *v*. A linear amplification is observed in the considered range of *v*. The slope of the linear fitting line of the dynamic stress is gentler than that of the acceleration, thereby indicating that the increment of the vertical peak value of the dynamic stress is slower than that of the acceleration. For example, as *v* increases from 60 to 160 km/h, the acceleration increases by 271%, whereas the dynamic stress increases by 18%. The amplification of the vertical peak of the acceleration with v is more evident than that for the vertical peak of the dynamic stress. The vertical peak values of both the acceleration and dynamic stress increase with *v* in a linear manner for the range of tested speeds, and the respective relations can be expressed as follows:

$$A_{rs,\max} = 0.083v - 2.1 \; R^2 = 0.999 \tag{7}$$

$$\sigma_{rs,\max} = 0.26v + 110.6 \; R^2 = 0.934 \tag{8}$$

where $A_{rs,max}$ and $\sigma_{rs,max}$, respectively, denote the vertical peak values of the acceleration (unit: m/s$^2$) and dynamic stress (unit: kPa) on the top layer surface; *v* represents the train speed (unit: km/h); and $R^2$ is the correlation coefficient.

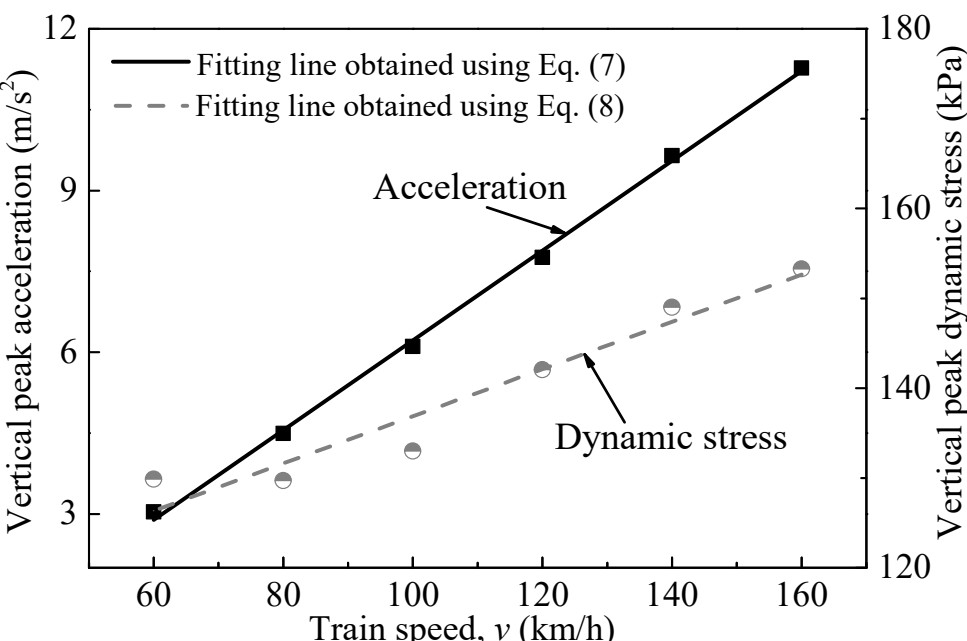

**Figure 9.** Variation of vertical peak acceleration and dynamic stress at the center of the top layer surface with the train speed.

### 3.3.2. Effect of Axle Load

Figure 10 shows the calculated variation of the vertical peak values of the acceleration and dynamic stress on the top surface of a subgrade bed with $F$ varying from 200 to 300 kN. With an increase in $F$, the vertical peak values of the acceleration and dynamic stress exhibit different linearly upward tendencies, which can be fitted based on the calculated data as follows:

$$A_{rs,\max} = 0.0152F \; R^2 = 1 \tag{9}$$

$$\sigma_{rs,\max} = 0.649F + 0.021 \; R^2 = 1 \tag{10}$$

where $F$ represents the axle load (unit: kN). The abovementioned linear fitting formulations indicate that $A_{rs,max}$ and $\sigma_{rs,max}$ increase with the increasing $F$. Furthermore, the increment in the $A_{max}$ is identical to that of the peak dynamic stress. For example, as $F$ increases from 200 to 300 kN, both the acceleration and dynamic stress increase by 50%. The amplifications for higher axle loads can also be inferred from these results as following a trend similar to the material response; however, only a limited range of axle loads have been considered in this study. This linear relationship is related to the fact that the kinematic energy transmitted to the system, which increases with the mass of the axle is proportional to the vertical peak values of the acceleration and dynamic stress.

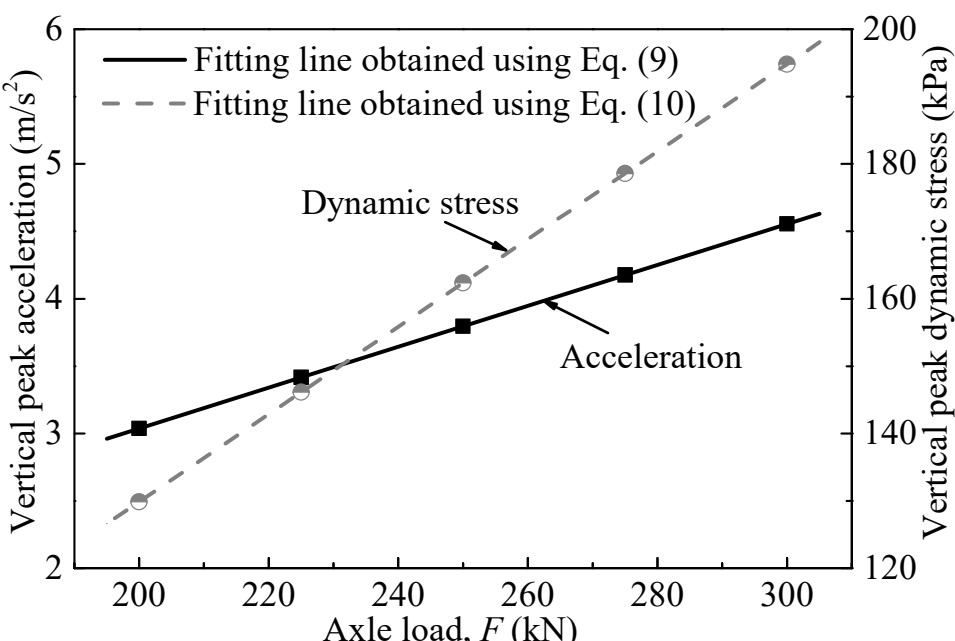

**Figure 10.** Variation of vertical peak acceleration and dynamic stress at the center of the top layer surface with the axle load.

### 3.3.3. Dynamic Response Attenuation in Soil

The $A_{max}$ commonly decreases with distance from source, and it seems that this general rule can also be found in the responses of track bed as shown in Figure 11. The acceleration levels along measurement line show significant decrease with increasing distance. The $A_{max}$ commonly decreases with distance from source, and only approximately 30–56% remains after attenuation for a subgrade bed depth of 2.5 m, whose value can be remained around 5% after attenuation of 7.0 m subgrade. Furthermore, $A_{max}$ tends to be uniform in the embankment layer at different $v$ and $F$, as the $A_{max}$ values are nearly coincident at a soil depth of more than 7.0 m. Additionally, Figure 11 also indicates that $v$ and $F$ have significant effects on the Amax on the top layer surface. It increases with $v$ and $F$. However, this effect varies with depth from the source. The closer to the source, the greater the impact.

In this work, $A_{max}$ is normalized by the corresponding values obtained on the top layer surface and expressed as an attenuation ratio of vibration. As the attenuation ratio deviation between the different conditions is low, all the cases can be averaged, as shown in Figure 11b. A best-fit curve can exhibit the variation of the attenuation ratio profile with increase in the depth, which can be expressed as an exponential function:

$$f_{AH} = 1.07e^{-0.29H} - 0.07 \ R^2 = 0.996 \tag{11}$$

where $f_{AH}$ is the attenuation ratio of the acceleration, $e$ is the natural logarithm constant, and $H$ is the distance to the bottom of the ballast along the track centerline.

Figure 12a shows the vertical peak dynamic stress distribution ($\sigma_{d,max}$) against the depth, as generated by moving trains for different v and F. The train loads are translated through the top and bottom layers to the embankment in a wave form. In the process of translation, a certain amount of energy is absorbed by damping, and the translating area becomes wider. Similar to the attenuation characteristics of the acceleration, $\sigma_{d,max}$ attenuates with an increase in the depth. However, Figure 12a indicates that $\sigma_{d,max}$ attenuates considerably faster than Amax does. For example, at the bottom of the top layer (0.6 m), $\sigma_{d,max}$ is attenuated by approximately 75%, whereas the attenuation is approximately 95% at the bottom of the bottom layer (2.5 m). $\sigma_{d,max}$ is attenuated by nearly 98% at the foundation (8.1 m), resulting in a negligible value compared to the weight of the subgrade filling.

A large proportion of $\sigma_{d,max}$ is visibly attenuated in the top and bottom layers, and this result is consistent with that reported in existing literature [35].

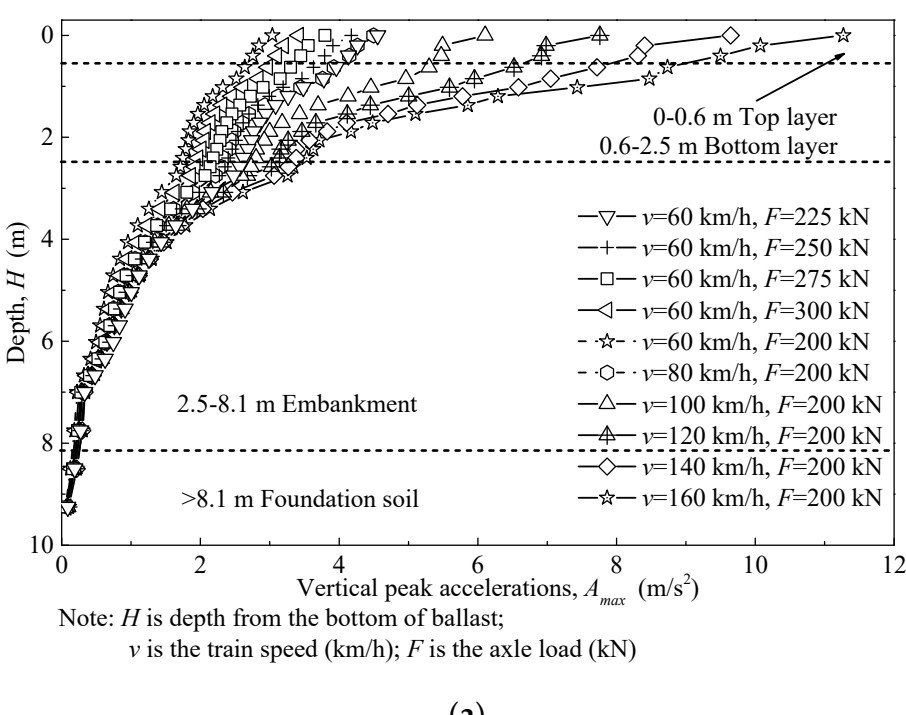

(**a**)

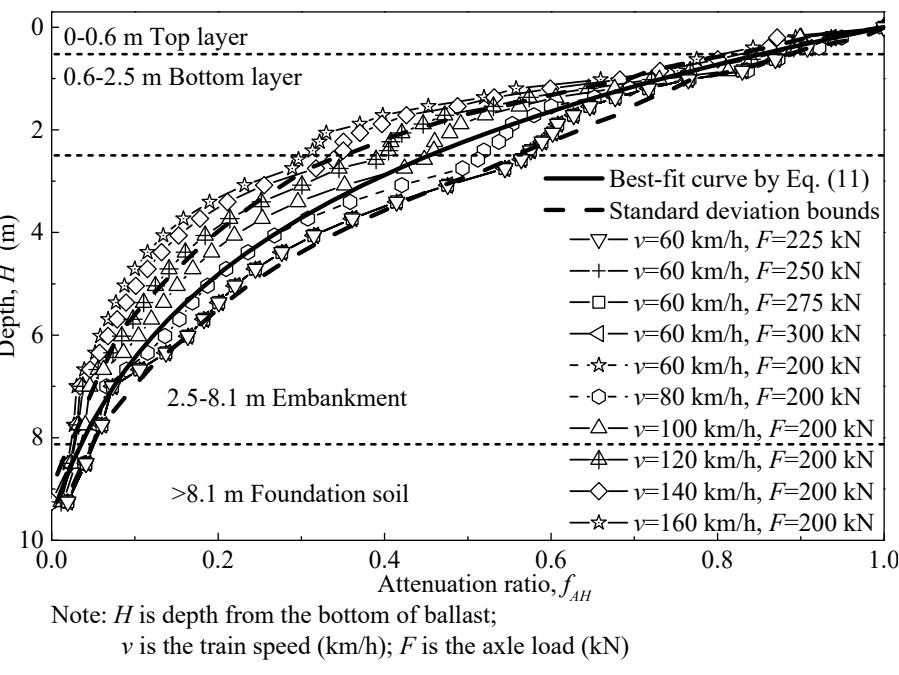

(**b**)

**Figure 11.** Acceleration distribution along the depth in the centerline for (**a**) vertical peak accelerations, $A_{max}$, and (**b**) attenuation ratio, $f_{AH}$.

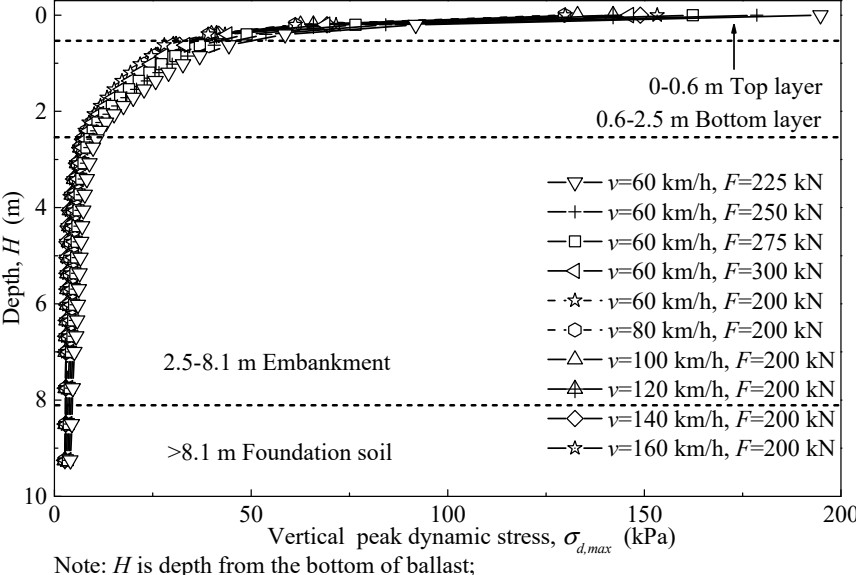

**(a)**

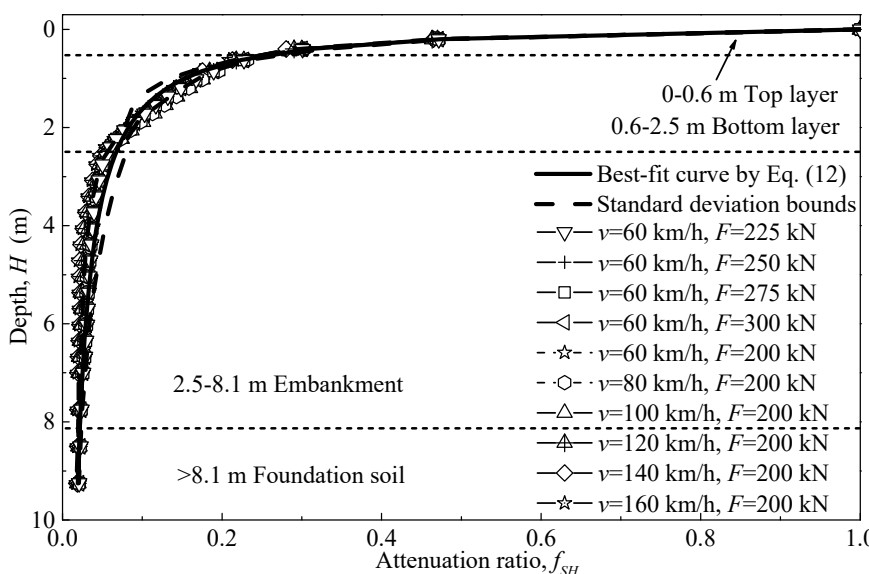

**(b)**

**Figure 12.** Acceleration distribution along the depth in the centerline for (**a**) vertical peak dynamic stress $\sigma_{d,\,max}$ and (**b**) attenuation ratio ($f_{SH}$).

The results of normalization of $\sigma_{d,max}$ at different depths using $\sigma_{d,max}$ at the top layer surface are shown in Figure 12b. It can be noted that the data can be fitted well by using the proposed empirical formula, that is Equation (12), which indicates that the vertical attenuation laws of attenuation for different F are similar.

$$f_{SH} = \frac{0.18}{0.18 + H} \tag{12}$$

where $f_{SH}$ is the attenuation ratio of the dynamic stress.

## 4. Evaluation of Service Performance

The dynamic stresses on the track structure, as induced by HAL trains, represent a critical factor that must be considered in the design of HAL railways. In this work, an integrated analysis of all the above mentioned numerical and empirical results was performed to evaluate the service performance. As mentioned previously, the dynamic stresses on the top layer surface are influenced by $v$ and $F$, and this influence can be expressed using Equations (8) and (10), respectively. Considering the vertical peak dynamic stress on the top layer influenced by the two factors presented above, a function of the variation in $v$ and $F$ is stated:

$$\sigma_{d,\max,v-F} = \sigma_0 \times f(v) \times f(F) \tag{13}$$

where $\sigma_0$ is a model parameter, and $f(v)$ and $f(F)$ depend on $v$ and $F$, respectively. Based on Equations (8) and (10), a nonlinear regression analysis of these numerical results in the following equation:

$$\sigma_{d,\max,v-F} = 0.0078 \times (0.26v + 110.6) \times (0.649F + 0.021) \tag{14}$$

By combining Equations (12) and (14), the distribution of $\sigma_{d,max}$ under the track structure with varying $v$ and $F$ can be predicted.

Figure 13 shows the ratio of $\sigma_{d,max}$ to the self-weight stress ($\sigma_s$) at the bottom layer (2.5 m below the top layer surface) with varying $v$ and $F$ for the locations directly below the track centerline. According to the Code for the Design of Heavy Haul Railway [36] of China, $\sigma_{d,max}/\sigma_s$ should be less than 0.20 to maintain the long-term stability of a railway subgrade. Consequently, the thickness of the top and bottom layers considered in the existing HAL railway network can likely satisfy the requirements to realize a heavy-haul train with a certain v and F within the available zone (see gray zone of Figure 13); however, the operation of HAL trains with v and F outside the available zone cannot be realized unless the top and bottom layers are reinforced. It should be noted that the operation of HAL trains with different $v$ and $F$ for the existing railway lines [7,25,37,38] lies in the predicted available zone (see Figure 13), which demonstrates that the proposed service performance evaluation method is valid and accurate.

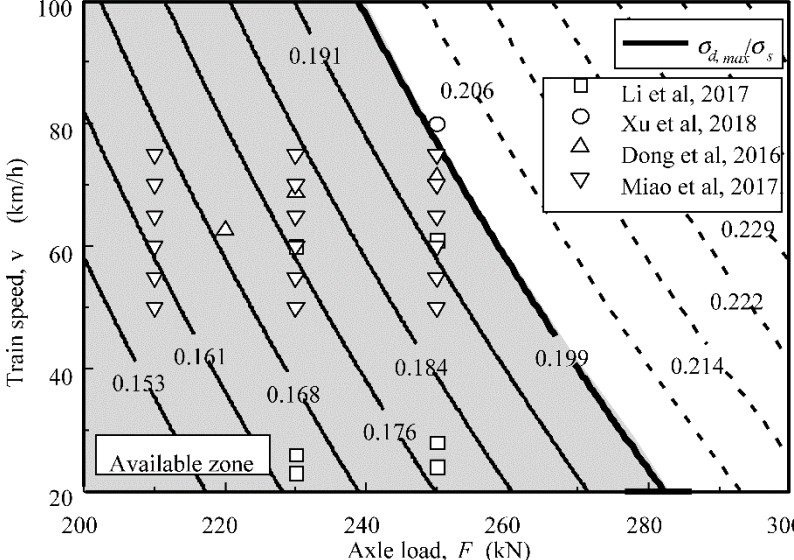

**Figure 13.** Effects of train speed ($v$) and axle load ($F$) on long-term stability of the existing heavy-haul railway system [7,25,35,36].

## 5. Conclusions

This paper proposes the use of LFWD-based inverse analysis to estimate the resilient modulus of the track substructure of the existing HAL railway systems. The resilient modulus for each layer of the substructure are set to the elastic parameters of a 3D FE model to perform a dynamic response analysis. In addition, a parametric study is performed to investigate the effects of $v$ and $F$ on the track characteristics. Accordingly, based on the numerical results, the necessary conditions to ensure long-term stability of the existing HAL railway are discussed. The following conclusions can be drawn from this work:

(1) The LFWD, which has been proven to be a versatile and reliable non-destructive testing device for road pavements, can be a valuable tool to realize the evaluation of China's HAL railway system. An agreement is noted between the 3D FE model and the deflection data, which further demonstrates that the proposed approach can be used to determine the resilient modulus with a satisfactory accuracy.

(2) The effects of various factors on the dynamic response are notable. The vertical peak acceleration ($A_{max}$) at the bottom layer reduced by approximately 30–50% compared to that at the top layer, whereas the dynamic stress reduced by approximately 95%. Furthermore, the $A_{max}$ and dynamic stress decayed with the depth exponentially and hyperbolically, respectively.

(3) The method of service performance evaluation is established based on the results of the parametric study on the effects of varying $v$ and $F$. Based on the recommendation that the critical stress ratio should be less 0.2, an endurance limit of $v$ and $F$ is assigned to ensure the long-term stability of the existing HAL railways. It was noted that $F$ can be up to 260 kN, whereas $v$ must be less than 60 km/h. However, if $v$ needs to be increased, the corresponding $F$ must be reduced to ensure service safety.

It must be noted that the findings in this work were derived considering the case of the HAL train C80, running between Baotou and Shenmu. It is expected that further analogous parametric studies performed considering trains with different configurations or subgrades with various resilient moduli, may lead to different results, which can help identify the different critical regions to improve the current study. Additionally, future research will need to be carried out to examine the vibration velocity of the subgrade which could not be achieved in this study due to the limitation of the current testing apparatus. Furthermore, various multi-body vehicle models will also be used to model the train load.

**Author Contributions:** Conceptualization and methodology, S.T. and X.L.; software, S.T.; validation, T.L. and I.-R.G.; writing—original draft preparation, S.T.; writing—review and editing, A.C.; funding acquisition, T.L. All authors have read and agreed to the published version of the manuscript.

**Funding:** This research was funded by the National Key Research and Development Project of China (grant no. 2018YFC1505305), the National Natural Science Foundation of China (grant no. 42102311), the State Key Laboratory for GeoMechanics and Deep Underground Engineering, China University of Mining & Technology (grant number: SKLGDUEK2103), the fellowship of China Postdoctoral Science Foundation (grant no. 2021M701014), and the Postdoctoral Fellowship of Heilongjiang Province (grant no. LBH-Z21062).

**Institutional Review Board Statement:** Not applicable.

**Informed Consent Statement:** Not applicable.

**Data Availability Statement:** Not applicable.

**Conflicts of Interest:** The authors declare no conflict of interest.

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
