# Peer review of "Evaluating the Service Performance of Heavy Axle Load Ballasted Railway by Using Numerical Simulation Method"

_applsci, doi:10.3390/app12052539_

Round 1

Reviewer 1 Report

The problem is interesting and is directly related to issues of safety and reliability of railway tracks. For many readers, the article will be interesting. The abstract is eloquent and there is no need to change anything about it. The introduction provides a good overview of the state of the art. It appropriately clarifies the background of the problem. I consider the method of experimental verification of the subsoil parameters to be correct and maximally necessary. Without experimentally validated data, numerical analysis would only be fictitious. The FEM model of the track is good. It is a pity that the authors model the load in the form of moving constant forces. Today, various multi-body vehicle models are already used. The frequency range of the 0-25 Hz load is not in accordance with the experiment. The real load also contains higher frequency components than 25 Hz with a dominant position. The model does not consider the rail. It is assumed that the wheel force acts on the sleeper in full. It's not true. The load is spread over 10-14 sleepers, depending on the rail fastening.  No more than 45% of the wheel power will be transferred to the sleeper. The agreement between the calculation and the experiment presented in FIG. 7 corresponds to the quality of the model. Better could not be achieved in this case. The effort to linearize the dependence of dynamic quantities on the speed of the vehicle I do not consider to be correct. Each such dependence has a large number of local maxima and spikes. If you were counting on a finer step of speed, you would be convinced of that. It only makes sense to talk about the influence of all other parameters in connection with the specific speed of the vehicle. I would rather avoid the statement about linear dependence in the conclusions (2). I recommend publishing the article so that more advanced models could be compared with less perfect ones in the future.

Reviewer 2 Report

The authors study the effects of train speed and axle load on the integrity of track substructure using a 3D numerical simulation. LFWD measurements were used to iteratively update the numerical models. Although the study discusses an interesting topic and is of interest to the railway community, several citations throughout the manuscript (section 3 in particular) made it less clear the actual novelty and contribution of the study. Other comments provided below are rather minor.

Line 66-68: “Therefore, it is … trains.” is a bit vague. Please rewrite this sentence.     

Line 72: What “UMAT” stands for?

Line 93: The term “dn” seems to refer to vertical displacement. If so, please clarify that in the manuscript.

Line 99: It does not seem Figure 2a was described in the manuscript. Please discuss this figure and elaborate on the nonuniform distribution of geophone sensors from the LFWD.

Line 129: Kindly label the layers in Figure 3. Also, it is more convenient to use x, y, z when referring to directions than longitudinal, horizontal, and vertical. Authors may keep using directions but should at least include coordinates as longitudinal (y), horizontal (x), etc.   

Line 172: Shouldn’t be Eq. (1) instead of Eq. (5)?

Line 182: Eq. (4) missing parenthesizes.

Line 245, and Line 323: coefficients (an, bn) need to be properly formatted. Please review the manuscript for similar issues.

Line 263: Figure 7 only indicates similarity in magnitude.  Could the author elaborate more on the similarity between predicted and actual responses in timing and shape?

Line 32-328: In which figure this finding was observed “Amax tends to be uniform…”? Figure 11a apparently shows this behavior over the lower segment of embankment. Please explain the relationship between depth and acceleration more clearly.  

Line 378: Please add one sentence to explain how Eq. (13) was derived.
